# The effects of fermented vegetable consumption on the composition of the intestinal microbiota and levels of inflammatory markers in women: A pilot and feasibility study

**Amy E. Galena[1], Jianmin Chai[2], Jiangchao Zhang[2], Michele Bednarzyk[3], Doreen Perez[3], Judith D. Ochrietor[4], Alireza Jahan-Mihan[1], Andrea Y. Arikawa[1]\***

1 Department of Nutrition and Dietetics, University of North Florida, Jacksonville, FL, United States of America, 2 Division of Agriculture, Department of Animal Science, University of Arkansas, Fayetteville, AR, United States of America, 3 School of Nursing, University of North Florida, Jacksonville, FL, United States of America, 4 Department of Biology, University of North Florida, Jacksonville, FL, United States of America

\* andrea.arikawa@unf.edu

**Data Availability Statement:** All relevant data are within the manuscript and its Supporting Information files.

## Abstract

The primary objective of this pilot study was to investigate the feasibility of regular consumption of fermented vegetables for six weeks on markers of inflammation and the composition of the gut microflora in women (clinical trials ID: NTC03407794). Thirty-one women were randomized into one of three groups: 100 g/day of fermented vegetables (group A), 100 g/day pickled vegetables (group B), or no vegetables (group C) for six weeks. Dietary intake was assessed by a food frequency questionnaire and blood and stool samples were provided before and after the intervention for measurement of C-reactive protein (CRP), tumor necrosis factor alpha (TNF-α), and lipopolysaccharide binding protein (LBP). Next-generation sequencing of the V4 region of the 16S rRNA gene was performed on the Illumina MiSeq platform. Participants' ages ranged between 18 and 69 years. Both groups A and B had a mean daily consumption of 91g of vegetables for 32 and 36 days, respectively. Serum CRP ranged between 0.9 and 265 ng/mL ($SD = 92.4$) at baseline, while TNF-α and LBP concentrations ranged between 0 and 9 pg/mL ($SD = 2.3$), and 7 and 29 µg/mL ($SD = 4.4$), respectively. There were no significant changes in levels of inflammatory markers among groups. At timepoint 2, group A showed an increase in *Faecalibacterium prausnitzii* ($P = 0.022$), a decrease in *Ruminococcus torques* ($P<0.05$), and a trend towards greater alpha diversity measured by the Shannon index ($P = 0.074$). The findings indicate that consumption of ~100 g/day of fermented vegetables for six weeks is feasible and may result in beneficial changes in the composition of the gut microbiota. Future trials should determine whether consumption of fermented vegetables is an effective strategy against gut dysbiosis.

**Funding:** AYA received an internal grant, the Us Women and Girls' Health Endowed Research Professorship, #764010. The funders had no role in study design, data collection and analysis, decision to publish, or preparation of the manuscript.

**Competing interests:** The authors have declared that no competing interests exist.

## Introduction

Diet significantly affects the gut microbiota throughout the lifespan of an individual [1–4]. It has been shown that *Bifidobacteria* predominate in the gut of infants receiving breast milk while formula fed infants have enriched amounts of *Bifidobacteria* and *Clostridia* [1]. Research indicates that a polysaccharide-rich diet such as a low-fat/high-fiber diet is correlated with an increased amount of Actinobacteria and Bacteroidetes and a decreased amount of Firmicutes [5, 6]. Furthermore, a Western-type diet, which is typically high in animal protein and fat and low in fiber, seems to be associated with lower abundance of beneficial bacteria such as *Bifidobacterium* and *Eubacterium* [6, 7]. Interestingly, findings from an elegant study conducted with humanized gnotobiotic mice indicated that consumption of a diet low in microbiota-accessible carbohydrates over four generations led to dysbiosis, characterized by inefficient transfer of low abundance taxa to the point of loss of almost 70% of the taxa present in the first generation [8].

Through a symbiotic relationship, gut microbiota play a fundamental role in the induction and function of the innate and adaptive immune system [9]. When dysbiosis occurs, the imbalance of commensal and pathogenic bacteria leads to the production of microbial antigens and metabolites, such as lipopolysaccharide (LPS) and cytokines that activate intestinal macrophages [10]. LPS is a component of the outer membrane of Gram-negative bacteria that induces inflammatory responses [11–13]. In humans, LPS is transported by LPS-binding protein (LPB), which is an acute phase protein synthesized in the liver to mitigate the biological actions of LPS [14–16]. Previous research suggests that changes in the profile of the gut bacteria may reduce levels of LPS and LPB [17–19]. Another inflammatory marker that has been significantly associated with dysbiosis is C-reactive protein (CRP) [20, 21]. A recent review by Munckhof et al [21] reported that the abundance of gut bacteria such as *Bifidobacterium*, *Faecalibacterium*, *Ruminococcus*, and *Prevotella* was inversely related to the inflammatory markers CRP and IL-6, demonstrating the importance of bacterial changes in the microbiome for the modulation of systemic inflammation.

Several dietary approaches have been linked to changes in abundance and diversity of specific microbial taxa [2, 6, 22–24]. In a 4-week longitudinal study, small but significant differences in overall microbial communities were found between consumers of fermented plants and non-consumers [25]. A recent clinical trial found that individuals randomized to a high-fermented foods diet of six or more servings per day showed significant increases in alpha diversity and improvements in inflammatory markers over a 10-week period [26], but more studies are needed to identify specific health benefits of various fermented foods. Fermented vegetables are both a source of prebiotics, due to their high content of plant polysaccharides and, probiotics [27, 28], such as *Lactobacillus brevis*, *Lactobacillus plantarum*, and *Leuconostoc mesenteroides* [29–32], which would constitute an ideal food to promote intestinal and metabolic health. In fact, one previous study found that consumption of 180 g of *kimchi* (fermented Napa cabbage) by obese Korean women led to an increase in relative abundance of the genus *Bifidobacterium* and a decrease in relative abundance of the genus *Blautia*, while no changes were seen in CRP levels [33]. Nielsen and colleagues [34] investigated the effects of daily lacto-fermented sauerkraut on irritable bowel syndrome symptoms of 34 Norwegian patients, of which 15 consumed a pasteurized sauerkraut supplement and 19 consumed an unpasteurized sauerkraut supplement for six weeks. In addition to improvement of symptoms, both groups also showed significant changes in gut microbiota composition with *Lactobacillus plantarum* and *Lactobacillus brevis* significantly elevated in the unpasteurized sauerkraut group [34].

In view of the scarcity of clinical studies investigating the role of fermented vegetables on inflammation and the gut microflora and considering that it is not known whether regular

consumption of fermented vegetables is a feasible dietary intervention for Western individuals who may not have been exposed to these types of vegetables, the primary aim of this pilot study was to assess the feasibility of regular consumption of 100 g of fermented vegetables for six weeks. Additional aims were to determine the effects of fermented vegetable consumption on markers of inflammation and the composition of the gut microflora.

## Materials and methods

This was a six-week, parallel arm, pilot and feasibility trial (clinical trial registration: NTC03407794). Female participants were randomly assigned to one of three treatment groups at an allocation ratio of 1:1:1. The treatment groups were: Group A (fermented vegetable group), Group B (pickled vegetable group), or Group C (control group). The duration of the trial was six weeks and data collection occurred primarily at the beginning and at the end of the six-week period.

### Study participants

The inclusion of only women was a stipulation of the funding source of this study. Additional inclusion criteria were: non-smoker, no previous diagnosis of cancer, no serious chronic disease, not on weight loss medication, not taking antibiotics at least three months prior to enrolling, not consuming fermented vegetables on a regular basis, not taking monoamine oxidase inhibitors, willing to consume fermented vegetables for six weeks, willing to be randomized to any treatment group. Exclusion criteria were uncontrolled hypertension, frequent use of antibiotics or probiotics, smoker, taking anti-inflammatory medication on a regular basis, having an auto-immune disease and age below 18 or above 70 years. This project was approved by the University Institutional Review Board (IRB#10334264) and all participants provided informed consent prior to starting the study.

### Study procedures

Study participants were recruited between January and October 2019, via flyers posted throughout commercial businesses in the metropolitan area of a northeastern Florida city, emails sent out to University staff, and media advertisements. Potential participants were instructed to contact research staff via phone or email using the contact information provided in the recruitment materials. The research staff performed a screening interview with interested participants to further confirm eligibility criteria.

Participants with verified eligibility completed an in-person orientation session where the study procedures were explained in detail, the informed consent was reviewed and signed, and a baseline clinic visit was scheduled. Participants received a stool collection kit and were instructed to collect a stool sample within 24 hours of their two clinic visits. Coolers and ice packs were provided to help participants maintain the stool samples cold until the morning of the clinic visits. Randomization into one of the three treatment groups occurred immediately after all baseline data were collected.

### Treatment groups

Participants randomized into the control group (group C) were asked to follow their usual diet without any drastic changes. Those randomized into the fermented vegetable (group A) or the pickled vegetable group (group B) were asked to consume 0.5 cup of vegetables per day for six weeks, which was equivalent to 100 g of cabbage or 80 g of cucumbers. Participants received seven one-cup containers of fermented cabbage and/or cucumbers (group A) or pickled

cabbage and/or cucumbers (group B), respectively, every two weeks, until the end of the study. The fermented vegetables were purchased from a local producer while the pickled vegetables were purchased from a local grocery store, at the beginning of the study. While the vegetables in group A and group B had similar taste profiles, one major difference was the presence of lactic acid bacteria in the fermented vegetables but not in the pickled vegetables.

All participants randomized into groups A and B were asked to limit consumption of sodium due to the higher sodium content of the fermented and pickled vegetables.

## Data collection

**Surveys.** Participants completed online surveys to assess food intake, demographics, and prescription medication intake. The DHQ-III, a 135-item food frequency questionnaire designed by the National Cancer Institute [35], was used to assess the participants' dietary intake at baseline and follow-up. Dietary variables of interest were total calories (kcal), carbohydrate (g), protein (g), fat (g), fiber (g), glycemic load, and Healthy Eating Index (HEI), which is used as a measure of overall diet quality based on alignment of dietary components to the recommendations of the Dietary Guidelines for Americans [36].

All study participants were given a log to record their gastrointestinal function (frequency of defecation and consistency of stools) and side effects (bloating, diarrhea, constipation, and headache). Participants randomized to groups A and B were also asked to fill out a log about their daily compliance to the intervention. For each day of participation in the study, participants reported the amount of vegetable consumed in cups (0, 0.12, 0.25, 0.50, 0.75 or 1 cup).

**Clinical data.** Study staff members obtained participants' height, weight, and body composition at each clinic visit. A Detecto 439 (Webb City, Missouri) Eye Level Beam Physician Scale 400ib x 4oz with Height Rod was used to measure height in centimeters to the nearest 0.1 cm. Weight and percent body fat were measured by multifrequency bioelectrical impedance (InBody 570, Cerritos, CA.) Blood pressure was measured twice by a nurse using a sphygmomanometer. At each of the two clinic visits, skilled nurses collected blood in two 8-mL serum separator tubes via venipuncture of the antecubital vein. Blood tubes were left at room temperature for 30 minutes before centrifugation at 1400 rpm for 10 minutes. Serum was transferred to 1.5 mL cryogenic tubes in 1-mL aliquots and stool samples were transferred to 1.0 mL cryogenic tubes in 150-mg aliquots.

**Assessment of biomarkers.** C-Reactive Protein (CRP) and Tumor Necrosis Factor (TNF) alpha were measured in serum by commercial ELISA kits (Cat#DCRP00 for CRP and Cat#DTA00D for TNF alpha, R&D Systems, Minneapolis, MN). Lipopolysaccharide Binding Protein (LBP) was measured in serum by a Pierce LAL chromogenic endotoxin quantitation kit (Cat#88282, ThermoFisher Scientific, Waltham, MA).

**Assessment of the gut microflora.** DNA extraction and next-generation sequencing of the V4 region of the 16S rRNA gene were performed. DNA was extracted from the frozen stool samples with the DNeasy PowerLyzer PowerSoil Kit (Qiagen, Germantown, MD, USA) per the manufacturer's protocol. A NanoDrop One (Thermo Fisher Scientific, Madison, WI, USA) was used to measure DNA concentration. The DNA was diluted to 10 ng/μL. Amplicon PCR was performed on the V4 region of 16S rRNA using the forward (5′-GTGCCAGCMGCCGCGGTAA-3′) and reverse (5′-GGACTACHVGGGTWTCTAAT-3′) primers with specific adapter for each sample. The PCR (polymerase chain reaction) products were electrophoresed on 1% agarose gel to verify the size of amplicons followed by purification using the SequalPrep Normalization Plate Kit (Invitrogen, Carlsbad, CA, USA). The purified PCR amplicons were pooled together to generate the sequencing library. qPCR (quantitative PCR) was used to quantify the consolidated library using the Kappa Library Quantification Kit

(Roche, Indianapolis, IN, USA), and the quality of the library was determined by an Agilent 2100 Bioanalyzer (Agilent, Santa Clara, CA, USA). Sequencing was performed in a pair-end modality on the Illumina MiSeq platform rendering 2 x 150 bp paired-end sequences (Illumina, San Diego, CA, USA).

**Sample size, randomization, and blinding.**   The sample size for this pilot trial was determined on basis of previous *kimchi* feeding trials that had been conducted in Korea [33, 37, 38]. These trials had enrolled between 21 and 24 participants and reported significant findings related to metabolic [37, 38] and microbial data [33]. The sample size target for this study was 35 to 40 participants.

A simple randomization scheme was used to allocate participants into one of the three groups. The random allocation sequence was generated by the principal investigator, while participant enrollment and allocation into treatment groups were conducted by research staff. Those randomized into the vegetable groups were blinded to the type of vegetable consumed. Research staff conducting the biomarker and microbiome analysis were also blinded to treatment allocation by labeling all samples with unique four-digit identification numbers that could not be traced back to a master spreadsheet that contained randomization information.

## Statistical analysis

The primary goal of the data analysis was to assess the feasibility of the study and obtain data on variability of the measures for the design of future adequately powered studies. Feasibility was assessed by compliance to the interventions (groups A and B) and participants' reports on the overall tolerance of everyday consumption of the vegetables. Compliance was assessed by two measures, number of days in the study, and total amount of vegetables consumed. If the interval between the first study visit and the second study visit was within 42±3 days, compliance to study duration was assigned as 100%. Similarly, if the total amount of vegetables consumed was 4200±300g, compliance to vegetable consumption was assigned as 100%. Total amount of vegetables consumed was calculated by converting the reported data in cups from the daily logs into grams, where 1 cup was equivalent to 200 g and 1/8 cup was equivalent to 25 g. From these data, an overall compliance rate was calculated by averaging the two values. Tolerance to the vegetables was assessed by proportion of participants who reported experiencing side effects which included nausea, vomiting, headache, bloating, diarrhea, and abdominal pain.

Descriptive statistics were generated by cross-tabulation. Categorical data are reported as frequency and percentages and continuous data are reported as median and interquartile range, as well as 95% confidence intervals. Changes in biomarkers of inflammation were compared among groups using Kruskal-Wallis tests, while Wilcoxon signed-rank tests were used to compare within group changes in blood and clinical outcomes as well as microbial abundance. IBM SPSS (Statistical Package for the Social Sciences), v.26 (IBM Corp., Armonk, NY) was used for the above described analyses. A *P*-value lower than 0.05 was considered statistically significant.

For analysis of the microbiome data, Mothur software v1.39.1 [39, 40] was used following the MiSeq SOP, including steps for quality-filtering, alignment against a 16S reference database (SILVA v132), and clustering into operational taxonomic units (OTUs) with a pairwise 97% identity threshold. The OTUs were then classified using the Ribosomal Database Project database [41]. Mothur v1.39.1. was used to calculate alpha diversity (microbial diversity within each sample) and beta diversity (microbial diversity between samples) [40]. Alpha diversity was assessed via observed operational taxonomic units (OTUs) for microbial richness and the Shannon index for species richness and evenness [42, 43]. Two indices were also used to measure beta diversity. A principal component analysis (PCoA) was used to discover the percent

of variability and potential associations among the groups represented by the Bray-Curtis (measure of differences in taxa abundance between communities) and Jaccard index (taxa presence/absence). Associations were computed between frequencies of the components and the two PCoA axes. An analysis of similarity (ANOSIM) was used to evaluate whether gut microbiota and diet composition were significantly different among the groups [44]. Linear discriminant analysis (LDA) effect size (LEfSe) was used to identify specific bacterial features that were enriched between time points in each group at the OTU level. LDA score > 2 was used as the cut-off value for a significant effect size [44].

## Results

The flow of participants throughout the study is shown in Fig 1. Recruitment took place between January and October 2019 and the study was completed in December, 2019. Out of 85 potential participants who were screened prior to eligibility assessment, 34 participants were randomized into one of the three treatment groups and 31 completed the study. The most common reason for being excluded from the study was the presence of autoimmune disease or other chronic diseases, such as diabetes or heart disease. All study aims were assessed in at least nine participants from each group.

Baseline demographic and dietary characteristics of study participants by randomization group are shown in Table 1. Several baseline characteristics were balanced among the groups, with exception of age and some dietary variables. Participants in the control group (group C) were younger the participants in the other two groups. This difference was also reflected in the intake of calories, macronutrients, and fiber.

The overall nutrition facts for the pickled and fermented vegetables were similar with respect to macronutrients and sodium content (S1 Table). Analysis of the bacteria present in the study vegetables indicated that Firmicutes were the most abundant phylum in all vegetables, regardless of fermentation status (S1 and S2 Figs). On the other hand, the predominant genera present in the fermented vegetables corresponded to the lactic acid producing bacteria, such as *Lactobacillus*, *Leuconostoc*, and *Weissella*. In contrast, the predominant genera found in the pickled vegetables were *Bacillales* and *Paenibacillus*.

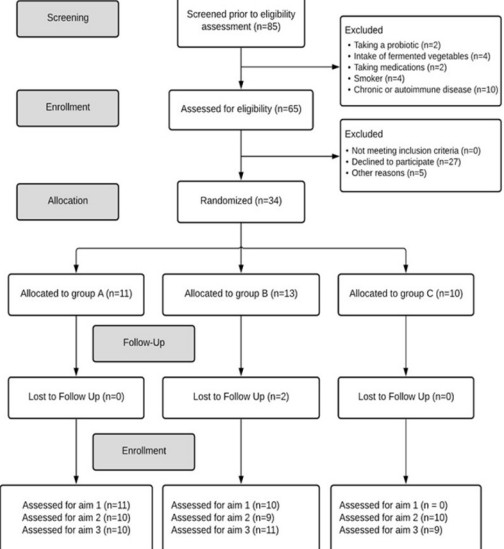

**Fig 1. Consort diagram showing the flow of participants in the fermented vegetable study.**

**Table 1. Demographic and dietary characteristics of study participants (*n* = 31).**

| Characteristics | Group A (*n* = 10) | Group B (*n* = 11) | Group C (*n* = 10) |
|---|---|---|---|
| Race | | | |
| Black | 1 (10%) | 1 (9.1%) | 1 (10%) |
| White | 8 (80%) | 9 (81.8%) | 6 (60%) |
| Other | 1 (10%) | 1 (9.1%) | 3 (30%) |
| Ethnicity | | | |
| Hispanic or LatinX | 1 (10%) | 2 (18.2%) | 2 (20%) |
| Non-Hispanic/LatinX | 9 (90%) | 9 (81.8%) | 8 (80%) |
| Education | | | |
| Some College | 2 (20%) | 2 (18.2%) | 2 (20%) |
| College Degree | 3 (30%) | 4 (36.4%) | 5 (50%) |
| Graduate Degree | 5 (50%) | 5 (45.4%) | 3 (30%) |
| Age (years) | | | |
| Mean (SD)[a] | 37.4 (13.9) | 39.2 (16.3) | 29.8 (11.4) |
| Median (IQR)[b] | 37 (44) | 44 (51) | 25 (31) |
| BMI (kg/m$^2$) | | | |
| Mean (SD) | 25.8 (6.9) | 26.1 (4.8) | 24.5 (3.8) |
| Median (IQR) | 22.7 (23.2) | 25.9 (18.5) | 22.9 (12.1) |
| Calories (kcal) | | | |
| Mean (SD) | 1460 (587) | 1488 (740) | 1671 (265) |
| Median (IQR) | 1334 (1929) | 1413 (2813) | 1623 (783) |
| Carbohydrate (g) | | | |
| Mean (SD) | 200 (81.5) | 186 (108.2) | 209.5 (61.7) |
| Median (IQR) | 174 (276) | 184.6 (387.1) | 215.9 (207.6) |
| Protein (g) | | | |
| Mean (SD) | 52.3 (28.5) | 61.3 (34.3) | 72.6 (12.2) |
| Median (IQR) | 48.9 (86.6) | 51.3 (118.8) | 54.7 (35.7) |
| Fat (g) | | | |
| Mean (SD) | 49.6 (25) | 55 (27.1) | 60.1 (13.3) |
| Median (IQR) | 48.9 (86.6) | 46.3 (97.5) | 54.7 (35.7) |
| Fiber (g) | | | |
| Mean (SD) | 22.2 (10.4) | 24.1 (16.9) | 18.9 (6.6) |
| Median (IQR) | 21.4 (32.2) | 19.7 (59.9) | 18.7 (20.8) |
| Glycemic load | | | |
| Mean (SD) | 146.7 (64.6) | 135.6 (87.9) | 161.9 (50.2) |
| Median (IQR) | 131.3 (212) | 139.7 (304.4) | 168.8 (181.4) |
| HEI-2015 Score[c] | | | |
| Mean (SD) | 71.6 (8.1) | 68.1 (12.5) | 65.9 (11.7) |
| Median (IQR) | 70.9 (24.6) | 71.1 (41) | 63.4 (34.8) |

[a]SD = Standard deviation.

[b]IQR = Interquartile range.

[c]HEI-2015 = Healthy Eating Index based on 2015 dietary guidelines.

Compliance and tolerance to the interventions are shown on Table 2. The mean number of days in the study for groups A, B and C were 39.4, 40.7 and 41.4 days, respectively. Group A consumed a total of 1609 grams of fermented cabbage and 1237 grams of fermented cucumbers, while group B consumed 1615 g of pickled cabbage and 1590 g of pickled cucumbers.

**Table 2. Reported compliance to the treatment groups, stool consistency and side effects by treatment group.**

|  | Group A | Group B | Group C |
|---|---|---|---|
|  | (*n* = 11) | (*n* = 10) | (*n* = 10) |
| Compliance (%) |  |  |  |
| Number of days—median (SD) | 81.1 (22.7) | 89.7 (11.6) | - |
| Amount of vegetables—mean (SD) | 77.1 (24.1) | 90.1 (12.1) | - |
| Overall, mean (SD) | 79.3 (21.8) | 89.9 (10.7) | - |
| Stool consistency[a] |  |  |  |
| Lumpy | 32 (18.3%) | 4 (3.7%) | 17 (13%) |
| Smooth and Soft | 135 (77.1%) | 74 (69.2%) | 94 (71.7%) |
| Fluffy | 8 (4.6%) | 17 (15.9%) | 20 (15.3%) |
| Watery | 0 (0) | 12 (11.2%) | 0 (0) |
| Side effects[b] |  |  |  |
| Bloating | 5 (45.5%) | 6 (60%) | 3 (30%) |
| Abdominal pain | 2 (18.2%) | 4 (40%) | 4 (40%) |
| Diarrhea | 1 (9%) | 3 (30%) | 1 (10%) |
| Nausea | 0 | 0 | 0 |
| Headache | 1 (9%) | 0 | 0 |

[a]Expressed as total number of days (%).

[b]Expressed as number of individuals (%) who reported experiencing each side effect at least 3 or more days during the 6-wk intervention period.

Overall compliance for group A was 79.3% and for group B, 89.9%. Bloating was the most common side effect reported by those in groups A and B, followed by abdominal pain. Notably, half of the participants in group A and 60% in group B experienced bloating during the study, compared with 30% of participants in the control group. On the other hand, abdominal pain was reported by 40% of participants in group B and group C, compared with 18% in group A. The most frequently reported stool consistency was soft and smooth in all three groups. Notably, fluffy stools were reported more frequently in groups B and C (over 15% of total days reported) compared with group A (4.6%).

Several inflammatory markers and other clinical outcomes were measured before and after the study intervention (Table 3). There were no significant differences in any of the parameters shown among the three groups. Pairwise comparisons within each group revealed that the control group (group C) had significantly lower percent body fat and systolic blood pressure at the end of the study. These changes were accompanied by significant decreases in intake of total calories and macronutrients in this group (S2 Table).

Alpha diversity of participants' stool samples is expressed as Shannon index, and the number of observed OTUs. Firmicutes, Actinobacteria, and Bacteroidetes represented the three predominant phyla in the stool samples of study participants. Firmicutes was the predominant phylum across all treatment groups with abundance ranging from 70 to 78% (Fig 2A and 2B). There were no significant differences within or between groups in relative abundance of the top phyla or top 20 genera shown in Fig 2.

Microbial diversity was measured through observed OTUs and Shannon index in this study. Box plots of the number of observed OTUs and Shannon index per treatment group and time point show thatat week 0, the number of observed OTUs for group C (C1) were significantly lower than for the group A (A1) and group B (B1), but no differences among groups were found at week 6. In contrast, there was a significant increase in Shannon index in group A (A2) compared with group C (C2), at week 6 (Fig 3). Fig 4 shows the top 20 abundant OTUs

**Table 3. Levels of inflammatory markers and other clinical parameters before and after the six-week intervention.**

| Clinical parameter [a] | Group A (n = 11) | Group B (n = 10) | Group C (n = 10) | P-trend[b] |
|---|---|---|---|---|
| BMI (kg/m$^2$) | | | | |
| Week 0 | 22.7 (7.3) | 26.1 (4.4) | 22.9 (6) | .594 |
| Week 6 | 23.3 (7) | 26.7 (4) | 22.8 (5) | .317 |
| P-value[c] | .058 | 0.964 | 0.443 | |
| Body fat (%) | | | | |
| Week 0 | 30.4 (22.6) | 36.7 (5.6) | 32.4 (12.8) | .769 |
| Week 6 | 31.4 (21) | 36.8 (6) | 31.1 (12) | .478 |
| P-value | .247 | .859 | .011 | |
| DBP (mmHg) | | | | |
| Week 0 | 81 (14) | 75.5 (16) | 75 (16) | .599 |
| Week 6 | 75 (17) | 72.5 (10) | 70 (13) | .241 |
| P-value | .476 | .389 | .374 | |
| SBP (mmHg) | | | | |
| Week 0 | 118 (18) | 110.5 (15) | 114 (23) | .804 |
| Week 6 | 121 (19) | 107 (18) | 104 (14) | .093 |
| P-value | .858 | .866 | .037 | |
| TNF-α (pg/mL) | | | | |
| Week 0 | 2.8 (4) | 4.5 (2) | 3.7 (3) | .378 |
| Week 6 | 2.6 (6) | 4.4 (2) | 3.1 (6) | .651 |
| P-value | .314 | .374 | .575 | |
| CRP (ng/mL) | | | | |
| Week 0 | 129.2 (308) | 209.2 (229) | 251.9 (1370) | .268 |
| Week 6 | 173.4 (375) | 211.4 (228) | 160.7 (746) | .772 |
| P-value | .214 | .086 | .139 | |
| LBP (µg/mL) | | | | |
| Week 0 | 13.3 (4) | 14.8 (6) | 12.8 (2) | .232 |
| Week 6 | 13 (5) | 12.7 (5) | 12.7 (7) | .621 |
| P-value | .508 | .066 | .721 | |

[a]Data are shown as median (IQR)

[b]P-trend for between group comparisons using the Kruskal-Wallis test.

[c]P-values represent within group comparisons using the Wilcoxon Singed-Rank test.

Abbreviations: BMI–body mass index; DBP–diastolic blood pressure; SBP–systolic blood pressure; TNF-α–tumor necrosis factor alpha; CRP–C-reactive protein; LBP–lipopolysaccharide binding protein.

per treatment group and time point. Wilcoxon Signed Rank tests were used to compare within group changes in OTUs 1 through 5. We found that OTU3 (*Faecalibacterium prausnitzii*) and OTU5 (*Roseburia faecis*) were significantly enriched at week 6 in group A (P = 0.022 and P = 0.037, respectively). No significant changes in these OTUs were found in groups B and C. Beta diversity was also investigated and PCoA plots based on Bray-Curtis and Jaccard distances are shown in Fig 5. Analyses using ANOSIM did not show any strong dissimilarities among or within groups for either Bray-Curtis or Jaccard distances.

LEfSe was used to identify specific bacterial features that were enriched between time points in each group at the OTU level (Fig 6). The results of the LEfSe analyses showed that OTU32 (*Ruminococcus torques*) was significantly less enriched at week 6 compared with week 0 in group A (Fig 6A). For group B, OTU206 (*Negativibacillus massiliensis*) was significantly more

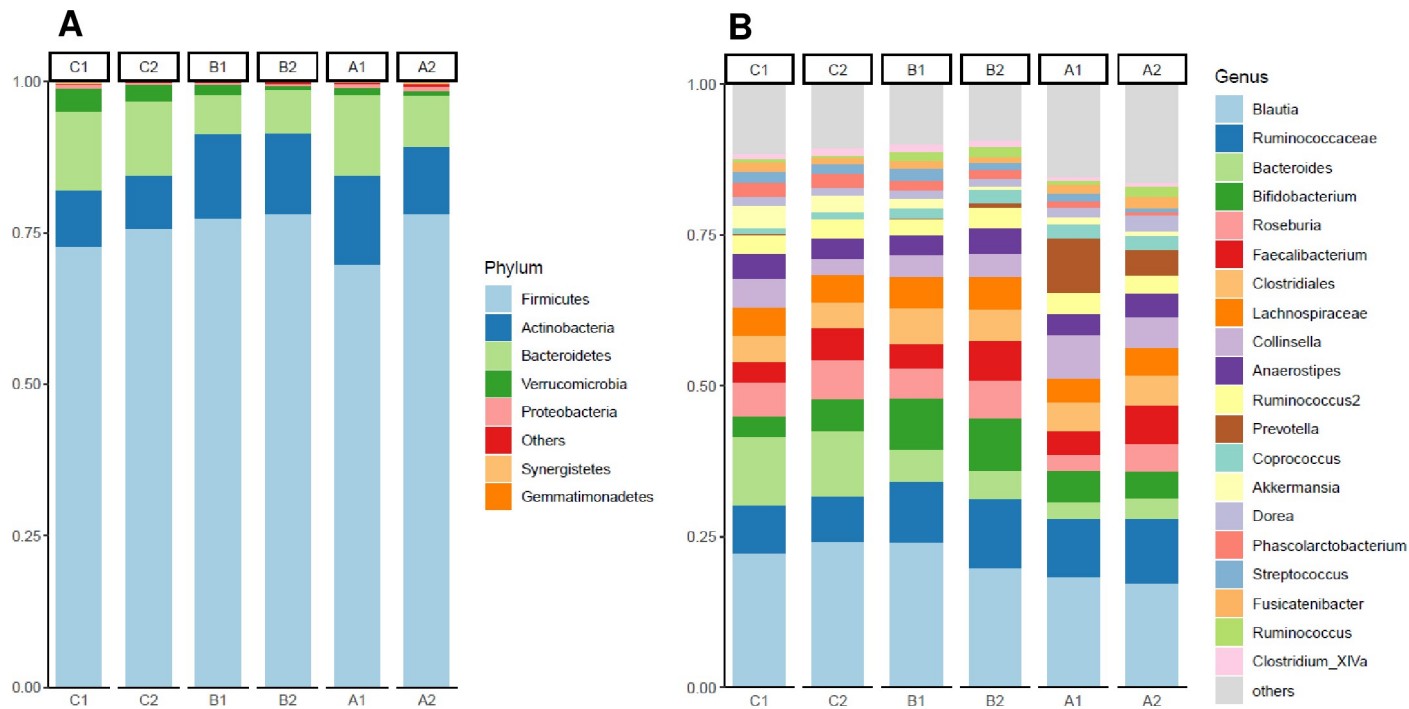

**Fig 2. Microbial composition ranked by relative abundance.** Observed phyla for treatment groups (A and B) or genus for individual participants (C and D) are shown. A1 = Group A (fermented vegetable) at week 0; A2 = Group A (fermented vegetable) at week 6; B1 = Group B (pickled vegetable) at week 0; B2 = Group B (pickled vegetable) at week 6; C1 = Group C (control) at week 0; C2 = Group C (control) at week 6.

enriched at week 6 than week 0 (Fig 6B) and for group C, OTU163 (*Mediterraneibacter glycyr-rhizinilyticus)*, was significantly less enriched at week 6 than week 0 (Fig 6C).

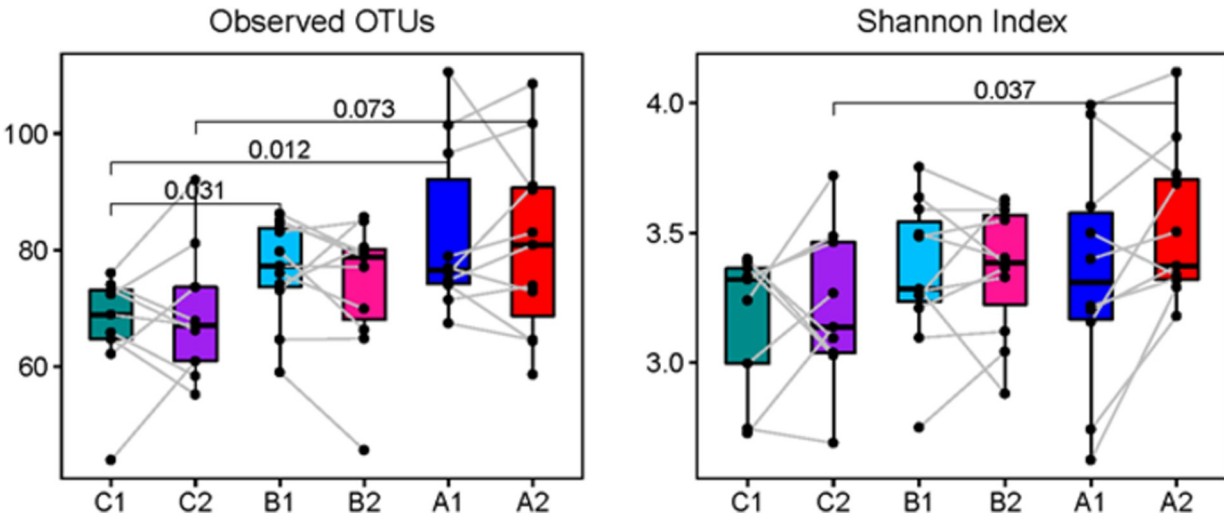

**Fig 3. Microbial diversity expressed as observed OTUs and Shannon index.** Box plots representing the observed OTUs and Shannon index are shown for each treatment group and time point. *P*-values for significant or nearly significant differences between groups are shown. The points with a connected line represent samples from the same individual at the two time points. A1 = Group A (fermented vegetable) at week 0; A2 = Group A (fermented vegetable) at week 6; B1 = Group B (pickled vegetable) at week 0; B2 = Group B (pickled vegetable) at week 6; C1 = Group C (control) at week 0; C2 = Group C (control) at week 6.

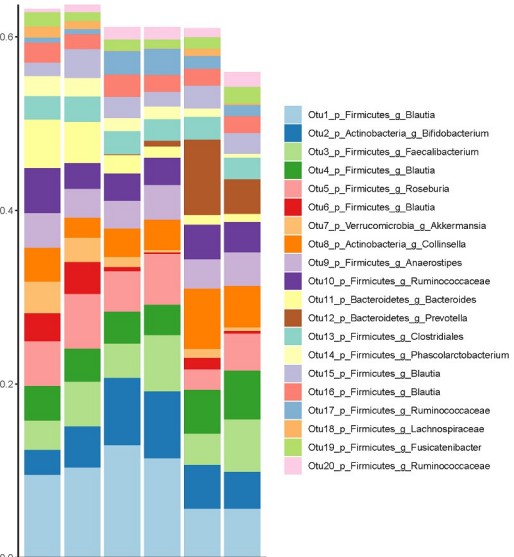

**Fig 4. Top 20 OTUs (operational taxonomic units).** OTUs are classified at the subgenus level and by relative abundance. OTU1 (*Blautia wexlerae*), OTU2 Bifidobacterium (*Bifidobacterium longum*), OTU3 Faecalibacterium (*Faecalibacterium prausnitzii*), OTU4 Blautia (*Blautia lut*), OTU 5 Roseburia (*Roseburia faecis*), OTU6 Blautia (*Blautia glucerasea*), OTU7 (*Akkermansia muciniphila*), OTU8 (*Collinsella aerofaciens*), OTU9 (*Anaerostipes hadrus*), OTU10 (*Ruminococcus bromii*). A1: fermented vegetable group timepoint 1, A2: fermented vegetable group timepoint 2, B1: non-fermented vegetable group timepoint 1, B2: non-fermented vegetable group timepoint 2, C1: control group timepoint 1, C2: control group timepoint 2).

## Discussion

This parallel arm pilot study explored the feasibility of regular consumption of fermented vegetables and their effects on markers of inflammation and the profile of the gut bacteria in a convenience sample of adult women living in the northeast region of Florida, United States.

Daily consumption of 0.5 cup (100 g) of fermented vegetables, such as sauerkraut and cucumbers for six weeks resulted in high compliance ranging from 79% in group A to 90% in group B. Nonetheless, several participants in both groups reported having difficulty consuming the vegetables every day towards the end of the study. Approximately 45% of group A participants and 60% of group B participants reported feeling bloated during the trial compared with 30% of group C participants. Although this side effect was expected with cabbage consumption, we found that several participants did not experience it, which may be related to differences in gut microbial composition. Based on these findings, some proposed amendments to the study protocol to improve compliance in a future trial would be to increase the variety of vegetables offered, to provide participants with a variety of recipes with ideas on how to incorporate the vegetables into their daily meals, and to increase the total duration of the trial while decreasing the frequency of consumption to four to five days per week.

Stool consistency was reported by participants in all treatment groups and the most frequently reported consistency was 'smooth and soft'. Interestingly, 'fluffy' stools, which may be indicative of increased intestinal transit time, were reported more frequently by participants in groups B and C than group A. This finding may indicate that consumption of fermented vegetables improved stool consistency of participants. Previous studies have shown that consumption of probiotics either as capsules [45] or as a drinking beverage [46] significantly improved stool consistency in individuals with constipation. It is difficult to directly compare findings from these studies with our findings, because presence of constipation was not an inclusion or

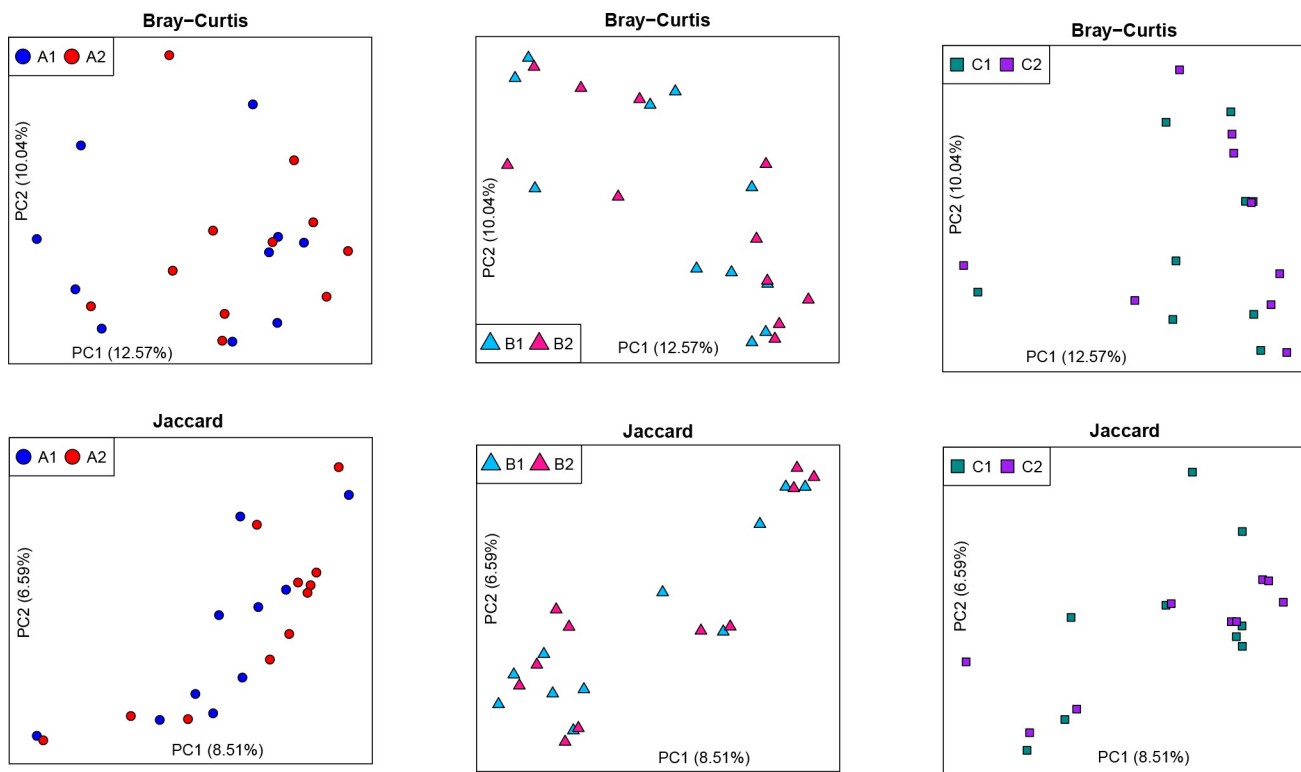

**Fig 5. Microbial β-diversity expressed as Bray-Curtis and Jaccard distances.** PCoA plots of Bray-Curtis and Jaccard distances are shown for each treatment. Week 0 and week 6 were compared in each. A1 = Group A (fermented vegetable) at week 0; A2 = Group A (fermented vegetable) at week 6; B1 = Group B (pickled vegetable) at week 0; B2 = Group B (pickled vegetable) at week 6; C1 = Group C (control) at week 0; C2 = Group C (control) at week 6.

exclusion criterion in our study. A more detailed assessment of constipation symptoms and stool consistency may provide additional insight into the role of fermented vegetables on digestive health.

One of the aims of this trial was to assess the role of fermented vegetables on markers of inflammation. Even though levels of CRP seemed to increase at week 6 for both groups A and B, there were no significant differences in the levels of inflammatory markers among groups or within groups. This was not surprising given the small sample size and the high between-person variability in the markers assessed. One previous study examining the effects of 180 g per day of fermented Chinese cabbage (kimchi) *versus* fresh Chinese cabbage on metabolic parameters of 24 obese Korean women for eight weeks found a trend towards increased CRP levels in the fermented kimchi group ($p = 0.052$) [33]. These findings were surprising and warrant additional trials to determine whether fermented vegetables can truly impact CRP levels.

Another aim of the study was to determine the gut microbiome composition in response to diets enriched in fermented or pickled vegetables. The main phyla identified in stool samples before and after the 6-week intervention were Firmicutes, with relative abundance of 75%, Actinobacteria (12%), Bacteroidetes (10%), Verrucomicrobia (1.8%), and Proteobacteria (0.5%). These findings support the results of a pilot trial conducted with Korean obese women, who's predominant phylum was Firmicutes, with relative abundance of 60 to 70%. On the other hand, unlike the present study, intake of 180 g of fermented kimchi significantly decreased the abundance of Firmicutes in the Korean women after eight weeks [33] while consumption of 100 g of fermented cabbage or cucumbers for six weeks did not result in lower

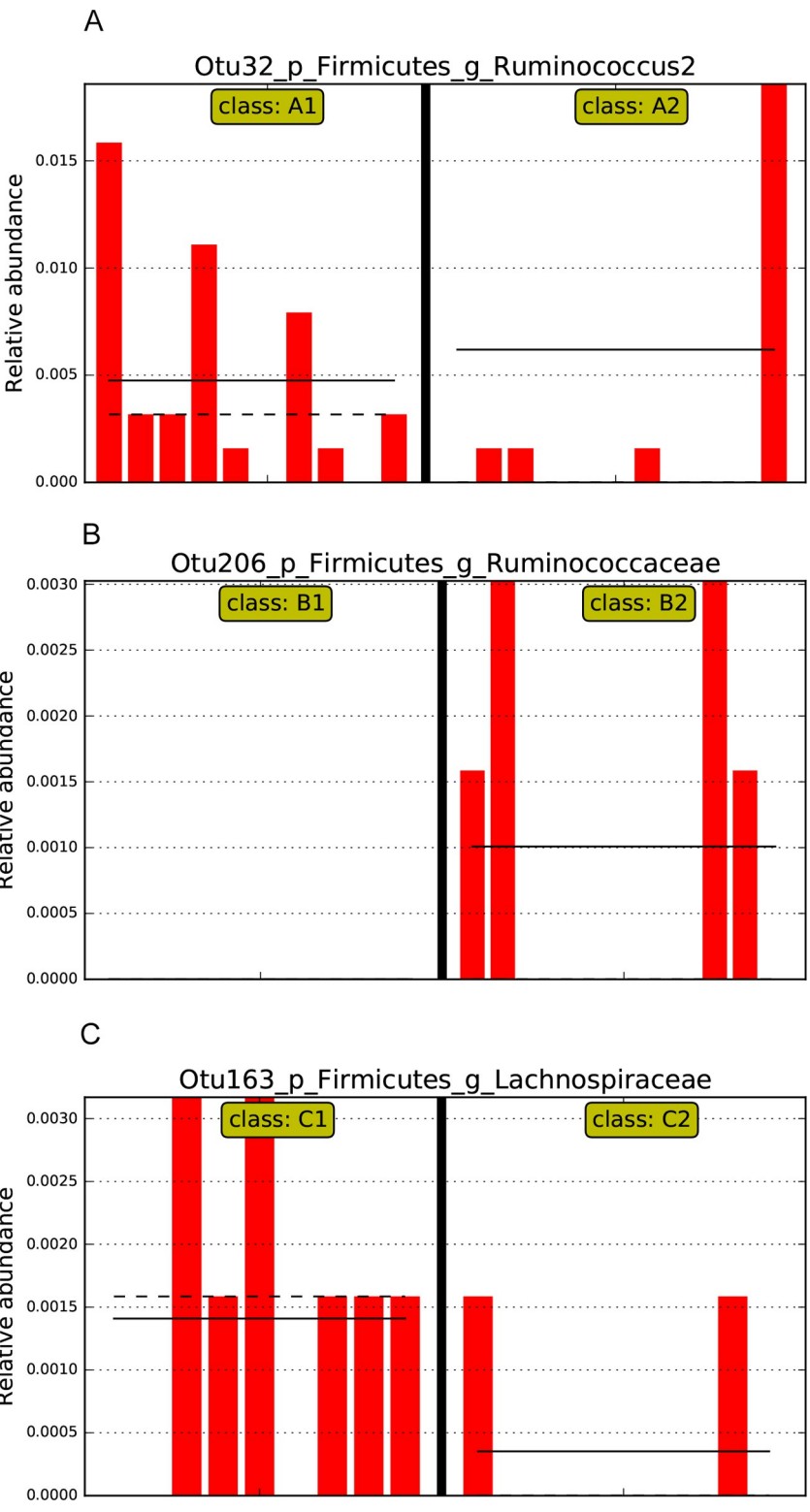

**Fig 6. LEfSe analysis of selected operational taxonomic units of individual participants.** Relative abundance of (A) OTU32 (*Ruminococcus torques*) for Group A, (B) OTU206 (*Negativibacillus massiliensis*) for Group B, and (C) OTU163 (*Mediterraneibacter glycyrrhizinilyticus)* for Group C are shown. Each bar represents one participant, and the order of participants is the same for week 0 and week 6. Solid horizontal lines represent mean relative abundance whereas the dashed horizontal lines represent the median relative abundance. A1 = Group A (fermented vegetable) at

week 0; A2 = Group A (fermented vegetable) at week 6; B1 = Group B (pickled vegetable) at week 0; B2 = Group B (pickled vegetable) at week 6; C1 = Group C (control) at week 0; C2 = Group C (control) at week 6.

abundance of this phylum. There are several possible reasons for these differences in results. First, the composition of the bacteria in kimchi used in the study by Han et al [33] may be different in comparison with sauerkraut. We were not expecting lower abundance of Firmicutes in the fermented vegetable group, considering that microbial analysis of these vegetables revealed high abundance of the genera *Lactobacillus* and *Leuconostoc*, *both belonging to the Firmicutes phylum*. Second, kimchi is prepared with many ingredients such as green onions, garlic and ginger, while none of these ingredients are present in sauerkraut, which could differently affect the growth of certain bacterial species [31]. Third, all participants in the Korean study had BMI above 27 kg/m$^2$, while participants in the present study had lower BMIs. It is possible that the differences in body fat may have influenced the effects of the fermented vegetables on the composition of the gut microflora. Despite the lack of significant changes in phyla composition in the present study, there was a significant increase in Shannon index in group A compared with group C, at week 6, suggesting that consumption of fermented cabbage may increase alpha diversity of the gut bacteria. It was also found that *Faecalibacterium prausnitzii* and *Roseburia faecis* were significantly enriched in group A at week 6 compared with week 0, but not in groups B or C. *Faecalibacterium prausnitzii* is a Gram-negative bacterium present in abundance in healthy individuals [47]. It has been associated with anti-inflammatory properties, protection of the intestinal barrier, oxidative stress tolerance [48] and inhibition of colonization of pathogenic bacteria [3]. Findings from a cell culture study indicate that *Lactobacillus* and other probiotic bacteria can stimulate the growth of *Faecalibacterium prausnitzii* [49], while a 12-month weight loss clinical study found that consumption of a Mediterranean diet significantly increased the abundance of *Faecalibacterium prausnitzii* and *Roseburia* [50]. *Roseburia faecis* is one of the most abundant commensals present in the large intestine and it plays a major role in fermentation of plant polysaccharides with production of short chain fatty acids, especially butyrate, which is an important nutrient for the colonic cells [51]. Previous research supports a beneficial role of the *Roseburia* genus, for instance, persons with obesity have lower abundance of *Roseburia* compared with normal weight individuals [52]. Similarly, patients with inflammatory bowel disease have significantly lower abundance of *Roseburia*, compared with healthy patients [53].

Results of the LEfSe analyses showed less abundance of *Ruminococcus torques* at week 6 compared to week 0, in group A. Research regarding the health effects of *Ruminococcus torques* is limited, but available data suggest an association of this species with adverse health outcomes [24, 54–58]. Meslier et al [24] showed lower abundance of *Ruminococcus torques* after an 8-week Mediterranean diet intervention compared to a control diet. Chatelier and colleagues [59] looked at associations between microbial richness and metabolic disease prevalence and concluded that *Ruminococcus torques* was a "potentially pro-inflammatory" species. Brahe and colleagues [57] reported positive correlations between *Ruminococcus torques* and insulin resistance and suggested the use of this bacterial species as a metabolic marker in postmenopausal women with obesity. Lastly, Odenwald and colleagues [58] also reported a positive association between *Ruminococcus torques* and insulin resistance, which was attributed to possible adverse effects on the gut barrier.

Most research on fermented vegetable intake has been conducted in Asian countries where fermented vegetables are widely consumed and in much larger quantities as compared to the typical consumption in the United States [31, 60–62]. A recent clinical trial, in which participants were randomized into either a high-fermented food diet or a high-fiber

diet, found that participants were able to increase fermented food consumption to six servings per day, most of which were achieved by increasing consumption of yogurt and vegetable brines [26]. Less is known about whether consumption of fermented vegetables is feasible in Westernized societies. Asian studies have used kimchi as the primary fermented food as compared to our study that used fermented sauerkraut and cucumbers, which contain different ingredients and different profiles of probiotic bacteria. The lack of research on this topic, particularly in the United States, leaves a gap in the knowledge about the health benefits of fermented vegetables for Western populations. One randomized double-blinded intervention conducted with 34 Norwegian patients suffering from irritable bowel syndrome found that, in addition to a decrease in disease symptoms, consumption of unpasteurized sauerkraut for six weeks led to an increase in detection of *Lactobacillus plantarum* and *Lactobacillus brevis* in subjects' stool samples [34]. We were not able to detect any changes in these two species in our study participants. These findings may have been related to the fact that several study participants chose fermented cucumbers as their vegetable of choice rather than sauerkraut. Analysis of the fermented vegetables indicated the presence of *Lactobacillus plantarum* but not *Lactobacillus brevis* in the fermented cucumbers while both species were present in fermented sauerkraut.

This pilot and feasibility study had a few limitations. The sample size was small, and the study participants represented a wide range of body weights and ages, which may have contributed to the baseline differences in the composition of the gut bacteria of the participants. Another limitation was the inclusion of only female participants in the study. Although this inclusion criterion was stipulated by the funding source for this project, the absence of male participants may have minimized the between-subject variations observed in the study. Regarding the implications of these findings to the design of future trials, it was determined that consumption of 100 g of fermented vegetables per day is feasible in women, but the inclusion of a variety of vegetables and/or flavors and a decrease in frequency of consumption with an increase in study duration might improve compliance. Future studies should also examine the effect of fermented vegetable consumption on the management of inflammatory disorders and other disorders in which inflammation plays an important role.

In conclusion, the findings from this pilot and feasibility study indicate that it is feasible for Western females to consume 0.5 cup of fermented cabbage and/or cucumbers every day for six weeks, noting that common side effects such as bloating may occur in some individuals. The effects of fermented vegetables on markers of inflammation and the gut microflora require further investigation. These data suggest that some positive changes in the abundance of certain bacterial species such as *Faecalibacterium prausnitzii* and *Ruminococcus torques* may be associated with consumption of fermented vegetables and future adequately powered human trials are necessary to unravel the relationship between fermented vegetables and health-related outcomes as well as microbial composition of the gut.

## Supporting information

**S1 Fig. Operational taxonomic unit (OTU) abundance of lactic acid bacteria genera in fermented and pickled cabbage and cucumbers provided in the study.**
(TIF)

**S2 Fig. Phyla and genera composition of bacteria present in the vegetables provided in the study.** BP = pickled cucumbers, BS = pickled cabbage, AP = fermented cucumbers, AS = fermented cabbage.
(TIF)

**S1 Table. Nutritional content of pickled and fermented vegetables provided in the study.**
(DOCX)

**S2 Table. Median values for selected dietary variables before and after the six-week intervention.** Group A: fermented vegetables, Group B: ficked vegetables, Group C: usual diet.
(DOCX)

**S1 Checklist. CONSORT 2010 checklist of information to include when reporting a pilot or feasibility trial**[*]**.**
(DOCX)

**S1 Protocol.**
(DOCX)

# Acknowledgments

The authors would like to acknowledge the study participants, the nurses who helped with collection of biological samples, and the undergraduate research assistants who helped with scheduling of appointments and data entry.

# Author Contributions

**Conceptualization:** Jiangchao Zhang, Judith D. Ochrietor, Andrea Y. Arikawa.

**Data curation:** Doreen Perez, Andrea Y. Arikawa.

**Formal analysis:** Jianmin Chai, Jiangchao Zhang.

**Funding acquisition:** Andrea Y. Arikawa.

**Investigation:** Amy E. Galena, Michele Bednarzyk, Andrea Y. Arikawa.

**Project administration:** Andrea Y. Arikawa.

**Resources:** Michele Bednarzyk, Doreen Perez.

**Supervision:** Andrea Y. Arikawa.

**Visualization:** Jianmin Chai, Jiangchao Zhang.

**Writing – original draft:** Amy E. Galena, Andrea Y. Arikawa.

**Writing – review & editing:** Amy E. Galena, Jianmin Chai, Jiangchao Zhang, Michele Bednarzyk, Doreen Perez, Judith D. Ochrietor, Alireza Jahan-Mihan, Andrea Y. Arikawa.

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
