## [Decision Letter · Decision Letter 0]

8 Feb 2022

PONE-D-21-29417

The effects of fermented vegetable consumption on the composition of the intestinal microbiota and levels of inflammatory markers in women: A pilot and feasibility study

PLOS ONE

Dear Dr. Arikawa,

Thank you for submitting your manuscript to PLOS ONE. After careful consideration, we feel that it has merit but does not fully meet PLOS ONE’s publication criteria as it currently stands. Therefore, we invite you to submit a revised version of the manuscript that addresses the points raised during the review process.

We look forward to receiving your revised manuscript.

Kind regards,

Jamie I. Baum, PhD

Academic Editor

PLOS ONE

https://journals.plos.org/plosone/s/file?id=ba62/PLOSOne_formatting_sample_title_authors_affiliations.pdf"

Reviewers' comments:

Reviewer's Responses to Questions

**Comments to the Author**

1. Is the manuscript technically sound, and do the data support the conclusions?

Reviewer #1: Partly

Reviewer #2: Yes

2. Has the statistical analysis been performed appropriately and rigorously? 

Reviewer #1: No

Reviewer #2: Yes

3. Have the authors made all data underlying the findings in their manuscript fully available?

Reviewer #1: Yes

Reviewer #2: Yes

4. Is the manuscript presented in an intelligible fashion and written in standard English?

Reviewer #1: Yes

Reviewer #2: Yes

5. Review Comments to the Author

Reviewer #1: The authors need to clarify and address these comments.

[Title] In this study, only cabbage or cucumber

[Abstract] Not clear “feasibility”? CRP, TNF-z, LBP in blood. What about analyzing the composition of gut microflora?

[Introduction]

Line 69-70. Need more details

Line 71-72. What are those high-fermented food diet?

* [Materials & Chemicals]

Line 127. How to control them not to have fermented vegetables during the study period?

Line 131-123. Need justification/rationale why cabbage or cucumber. Why are there two choices instead of one sample? What is the Nutrition Value of each vegetable? Different fiber (or other nutrients) content could affect the gut microbial composition.

Line 134-136. [1] Need more details of fermented cabbage/cucumbers and pickled ones- what ingredients are in it. Etc. [2] No data about taste profiles and bacterial composition.

Line 205 & 207. Why 3 days? Why 300g?

[Results]

Line 247. Fig 1. Need to mention about aim 1, 2, and 3

Table 1. Black-African or African America? White-Caucasian?

Need statistical analysis/differences for all tables

Table 2. “Overall, mean (SD)” – is it correct way to see compliance? Averaging those two?

Line 269-270. But can we know if they were experiencing it at first but got better? 3 days out of 42 days can be shot and also how do you make sure the side effect is from fermented/pickled samples, not from other food in their diet?

Line 273. Among the three groups � both before and after?

Line 322-324. Decreased in most individuals in group A at week 6 compared to week 0 except on individual with high increase

Line 326. 5C � any significant difference?

[Discussion]

Line 363. Better to show the abundance before and after to clarify the results

Line 365-366. Need a reference. With relative abundance of 60-70% was this data before or after the study?

Line 378 & 382. Just cabbage? Not cucumber?

Line 393. No discussion about Negativibacillus massiliensis and Mediterraneibacter glycyrrhizinilyticus

Line 410-411. if it can affect individuals’ results even in the same group, why was it allowed for participants to choose one of those?

Line 416. What about analyzing with incremental data? Then there might be a change?

* Editorial errors

Reviewer #2: A randomized pilot clinical trial was conducted which aimed to investigate the feasibility of regular consumption of fermented vegetables on inflammation and the composition of gut microflora. No significant changes in the levels of inflammatory markers were observed.

Minor revision:

Indicate the date range subjects were enrolled in the study.

6. PLOS authors have the option to publish the peer review history of their article (what does this mean?). If published, this will include your full peer review and any attached files.

Reviewer #1: No

Reviewer #2: No

---

## [Author Response · Author response to Decision Letter 0]

12 Feb 2022

All responses to the editor's and reviewers' comments were uploaded as a separate file titled Rebuttal Letter.

---

## [Decision Letter · Decision Letter 1]

1 Jul 2022

PONE-D-21-29417R1The effects of fermented cabbage and cucumber consumption on the composition of the intestinal microbiota and levels of inflammatory markers in women: A pilot and feasibility studyPLOS ONE

Dear Dr. Arikawa,

Thank you for submitting your revised manuscript to PLOS ONE. After careful consideration, we feel that it has merit but does not fully meet PLOS ONE’s publication criteria as it currently stands. Therefore, we invite you to submit a revised version of the manuscript that addresses the points raised during the review process.

Your manuscript has been assessed by an additional expert reviewer, whose comments are appended below. The reviewer has requested additional information about several aspects of the methodology and some additional data to fully contextualise your findings. Please ensure you respond to each point carefully in your response to reviewers document, and modify your manuscript accordingly.

We look forward to receiving your revised manuscript.

Kind regards,

Joseph Donlan

Editorial Office

PLOS ONE

Journal Requirements:

Reviewers' comments:

Reviewer's Responses to Questions

**Comments to the Author**

1. If the authors have adequately addressed your comments raised in a previous round of review and you feel that this manuscript is now acceptable for publication, you may indicate that here to bypass the “Comments to the Author” section, enter your conflict of interest statement in the “Confidential to Editor” section, and submit your "Accept" recommendation.

Reviewer #2: All comments have been addressed

Reviewer #3: (No Response)

2. Is the manuscript technically sound, and do the data support the conclusions?

Reviewer #2: (No Response)

Reviewer #3: Partly

3. Has the statistical analysis been performed appropriately and rigorously? 

Reviewer #2: (No Response)

Reviewer #3: Yes

4. Have the authors made all data underlying the findings in their manuscript fully available?

Reviewer #2: (No Response)

Reviewer #3: Yes

5. Is the manuscript presented in an intelligible fashion and written in standard English?

Reviewer #2: (No Response)

Reviewer #3: Yes

6. Review Comments to the Author

Reviewer #2: (No Response)

Reviewer #3: Overall, there are some necessary data that are missing. These data must be included and discussed in the paper or the paper should not be accepted. The data that this reviewer suggest are:

1. Characteristics fermented (pickle) cabbage and cucumber should be presented and discussed. For example, microbiota of fermented (pickle) cabbage and cucumber, besides their lactic acid bacteria, pH, SCFA, and others. These data are very important to support the change of gut microbiota composition

2. Data on faecal quality (pH and consistency) are very important to be presented and discussed as it shows the direct impact of the intervention. This can also be done by adding the presence of viable microorganisms in products of each intervention

3. Bloating (presented the intestinal disorder caused by diet intervention) must be recognized in the conclusion.

4. Line 326-327 - it was found that OTU3 (Faecalibacterium prausnitzii) was significantly enriched at week 6 in group A (P=0.022) – in this Figure 3, OTU3 does not clearly shows this bacterium has indeed significantly enriched in week 6. Since Faecalibacterium prausnitzii was mentioned in the conclusion as increased significantly, the data should be clear presented, as well as data on Ruminococcus torques

7. PLOS authors have the option to publish the peer review history of their article (what does this mean?). If published, this will include your full peer review and any attached files.

Reviewer #2: No

Reviewer #3: No

---

## [Author Response · Author response to Decision Letter 1]

5 Jul 2022

We appreciate the additional comments provided by reviewer 3 and we hope that the revisions are satisfactory. In response to reviewer 3 comments, we have significantly expanded some of the results and the discussion section. All revisions are highlighted in the rebuttal letter submitted. Sincerely, Andrea Arikawa.

---

## [Decision Letter · Decision Letter 2]

28 Jul 2022

PONE-D-21-29417R2The effects of fermented vegetable consumption on the composition of the intestinal microbiota and levels of inflammatory markers in women: A pilot and feasibility studyPLOS ONE

Dear Dr. Arikawa,

Thank you for submitting your manuscript to PLOS ONE. After careful consideration, we feel that it has merit but does not fully meet PLOS ONE’s publication criteria as it currently stands. Therefore, we invite you to submit a revised version of the manuscript that addresses the points raised during the review process. The reviewer has raised a small number of concerns that should be addressed. Once these changes have been made your manuscript will be ready for publication.

We look forward to receiving your revised manuscript.

Kind regards,

George Vousden

Staff Editor

PLOS ONE

Journal Requirements:

Reviewers' comments:

Reviewer's Responses to Questions

**Comments to the Author**

1. If the authors have adequately addressed your comments raised in a previous round of review and you feel that this manuscript is now acceptable for publication, you may indicate that here to bypass the “Comments to the Author” section, enter your conflict of interest statement in the “Confidential to Editor” section, and submit your "Accept" recommendation.

Reviewer #2: (No Response)

2. Is the manuscript technically sound, and do the data support the conclusions?

Reviewer #2: Yes

3. Has the statistical analysis been performed appropriately and rigorously? 

Reviewer #2: Yes

4. Have the authors made all data underlying the findings in their manuscript fully available?

Reviewer #2: Yes

5. Is the manuscript presented in an intelligible fashion and written in standard English?

Reviewer #2: Yes

6. Review Comments to the Author

Reviewer #2: Minor revision:

1- Line 233: The Mann-Whitney test is generally referred to as the Mann-Whitney U test.

2- Line 280: In addition to the means, include the corresponding standard deviations.

3- Line 304: Typographical error: Wilcoxon signed-rank test

4- Table 3: Label the last column, p-value instead of p-trend. Technically the Kruskal-Wallis tests for differences among groups rather than testing for a trend.

7. PLOS authors have the option to publish the peer review history of their article (what does this mean?). If published, this will include your full peer review and any attached files.

Reviewer #2: No

---

## [Author Response · Author response to Decision Letter 2]

31 Jul 2022

Reviewer #2: Minor revision:

1- Line 233: The Mann-Whitney test is generally referred to as the Mann-Whitney U test. There is no mention of the Mann-Whitney test in our manuscript as this test is primarily used for comparisons between two independent groups and we wanted to compare three independent groups.

2- Line 280: In addition to the means, include the corresponding standard deviations. Standard deviations were included

3- Line 304: Typographical error: Wilcoxon signed-rank test: this error was corrected 

4- Table 3: Label the last column, p-value instead of p-trend. Technically the Kruskal-Wallis tests for differences among groups rather than testing for a trend. Label was corrected

---

## [Editor Report · Decision Letter 3]

13 Sep 2022

The effects of fermented vegetable consumption on the composition of the intestinal microbiota and levels of inflammatory markers in women: A pilot and feasibility study

PONE-D-21-29417R3

Dear Dr. Arikawa,

We’re pleased to inform you that your manuscript has been judged scientifically suitable for publication and will be formally accepted for publication once it meets all outstanding technical requirements.

Kind regards,

George Vousden

Staff Editor

PLOS ONE
---

## [Editor Report · Acceptance letter]

19 Sep 2022

PONE-D-21-29417R3 

The effects of fermented vegetable consumption on the composition of the intestinal microbiota and levels of inflammatory markers in women: A pilot and feasibility study 

Dear Dr. Arikawa:

I'm pleased to inform you that your manuscript has been deemed suitable for publication in PLOS ONE. Congratulations! Your manuscript is now with our production department. 

Kind regards, 

on behalf of

Dr. George Vousden 

Staff Editor

PLOS ONE